# Gene synthesis allows biologists to source genes from farther away in the tree of life

Aditya M. Kunjapur [1,5], Philipp Pfingstag[2,6] & Neil C. Thompson [3,4]

Gene synthesis enables creation and modification of genetic sequences at an unprecedented pace, offering enormous potential for new biological functionality but also increasing the need for biosurveillance. In this paper, we introduce a bioinformatics technique for determining whether a gene is natural or synthetic based solely on nucleotide sequence. This technique, grounded in codon theory and machine learning, can correctly classify genes with 97.7% accuracy on a novel data set. We then classify ~19,000 unique genes from the Addgene non-profit plasmid repository to investigate whether natural and synthetic genes have differential use in heterologous expression. Phylogenetic analysis of distance between source and expression organisms reveals that researchers are using synthesis to source genes from more genetically-distant organisms, particularly for longer genes. We provide empirical evidence that gene synthesis is leading biologists to sample more broadly across the diversity of life, and we provide a foundational tool for the biosurveillance community.

[1] Department of Chemical Engineering, Massachusetts Institute of Technology, Cambridge, MA 02139, USA. [2] Sloan School of Management, Massachusetts Institute of Technology, Cambridge, MA 02142, USA. [3] Computer Science and Artificial Intelligence Lab, Massachusetts Institute of Technology, Cambridge, MA 02139, USA. [4] Laboratory for Innovation Science at Harvard, Harvard University, Cambridge, MA 02134, USA. [5]Present address: Department of Chemical and Biomolecular Engineering, University of Delaware, Newark, DE 19716, USA. [6]Present address: School of Management, Technical University of Munich, D-80333 Munich, Germany. These authors contributed equally: Aditya M. Kunjapur, Philipp Pfingstag. Correspondence and requests for materials should be addressed to A.M.K. (email: kunjapur@udel.edu) or to N.C.T. (email: neil_t@mit.edu)

Biologists and bioengineers often transfer genes across organisms to test genetic hypotheses or to endow their favorite model organisms with novel traits or functionality[1,2]. In the first industrial example of recombinant DNA technology, Eli Lilly and Genentech expressed a synthetic gene encoding human insulin in the model bacterium *Escherichia coli* for drug manufacturing[3]. Soon afterwards, biologists began sourcing genes encoding thermostable polymerases[4] from thermophilic bacteria and the well-known green fluorescent protein (GFP)[5] from the jellyfish as research tools. More recent biological research focused on mammalian models has featured considerable introduction of bacterial genes, notably the targeted genome editing tool CRISPR-Cas9[6–8] and tools for optogenetics[9,10]. The growing field of synthetic biology also drives gene transfer because the genome sequences of non-model organisms present a treasure trove of potentially novel and orthogonal genes for testing in model organisms[11,12].

Using DNA synthesis to transfer synthetic gene sequences from one organism to another may succeed where transferring natural gene sequences would fail. Although natural genes have the potential for direct transfer from one organism to another because of the universality of the genetic code, many such sequences would express poorly when moved into a new organism because of differences in codon usage, GC content, or the presence of expression-limiting regulatory elements[13,14]. These concerns only worsen as sequence length increases because the potential for problematic codons increases, as does the time required to manually convert these codons using PCR-based or restriction enzyme-based approaches. Such constraints can limit what genetic engineers accomplish.

In contrast with these restrictions on moving genes using traditional methods, gene synthesis can faithfully and rapidly recode natural sequences of large lengths[15,16]. Recoding algorithms harness synonymous codons that more closely reflect the expression organism and preserve the natural protein sequence[17]. Though the subtle implications of codon choice for the rate and quality of protein production are still being understood[18,19], such codon-optimization is so valuable for expression that commercial gene synthesis service providers typically offer this option by default. We posit that codon-optimization offers a promising way to identify synthetic genes and the engineered organisms that contain them and thus provides the first way, to the best of our knowledge, to identify synthetic sequences from sequence alone. In the past, such engineering efforts could have been detected through the scars from gene editing, but such methods are becoming obsolete because of advances in scar-less molecular cloning[20,21] and genome engineering techniques[22].

The ability to accurately identify synthetic genes enhances biosurveillance for organisms taking on non-native traits, which may be harmful or illicit. Although commercial DNA synthesis suppliers screen orders for similarity to select agents[23–26], detection of synthetic genes within organismal genomes is particularly valuable for cases where conventional biosecurity control could be circumvented, such as when synthesis is done on a non-regulated machine. Such detection is also relevant for biosafety in the event of accidental release of engineered organisms. The importance of additional biosurveillance capability has been articulated widely, for example by a major U.S. bipartisan biodefense study[27], ongoing U.S. intelligence agency research programs[28] and in agricultural contexts by the USDA Animal and Plant Health Inspection Service[29]. Furthermore, a June 2018 report commissioned by the U.S. National Academies identified that making existing bacteria more dangerous and in situ production of harmful biochemicals are two topics that warrant the most concern[30].

In addition to its biosecurity relevance, such classification could shed light on whether researchers in the life sciences are using synthetic genes differently than natural genes. We investigate these trends in the Addgene plasmid repository[31], which is a fruitful place to investigate these trends because it is a go-to repository for academic access to plasmids[32], as well as the most-used source for CRISPR-Cas9[33]. This prominence is evidenced in the rapid rise of the number of plasmids deposited over time (Fig. 1a), orders per year (Fig. 1b), and new labs depositing plasmids (Fig. 1c). Addgene is also used across a wide variety of expression platforms (Fig. 1d). Furthermore, the Addgene database has the distinct advantage of being publicly viewable (in contrast with the undoubtedly large, but proprietary databases in the biotechnology industry). Reliance on synthesis is likely to be most important for long sequences, where codon mismatches would be more challenging to address with traditional methods and for sequences transferred to more dissimilar organisms where the codon mismatch would be greatest[14]. At the same time, gene synthesis is usually more expensive than amplification of natural DNA, and synthesis cost, time, and error rates increase as sequence length increases[15]. Thus, we expect to see differential usage patterns in the Addgene data: with natural sequences used for shorter, genetically-proximate transfer and synthetic sequences used for longer, genetically-distant transfer. Ideally, our hypothesis on genetic distance could be tested directly using codon usage mismatch, but because codon usage tables are not available for many of the organisms that we study, we instead test our hypothesis using a correlate[34]: evolutionary genetic distance.

Based on the biological and engineering implications of synthesis, we postulate a set of features that have the potential to distinguish synthetic sequences. To discern which of these are most predictive, we construct two reference sample sets, each comprised of known synthetic and known natural sequences. Using the first of these, our training set, we evaluate the predictiveness of the features using machine learning techniques. Interestingly, some commonly-known distinguishers (e.g., rare codon content) provide no additional benefit for our predictor. Having decided on a predictor, we examine its predictiveness out-of-sample using our larger second reference sample or test set. We can correctly classify 97.7% of those sequences, confirming that our scalable, sequence-only method for detecting synthetic genes is highly effective. After analyzing ~19,000 Addgene sequences, we find that the average genetic distance between source and expression organism is greater for synthetic genes than natural genes and that this difference increases at longer sequence lengths. Our findings of how gene synthesis is being used in public repositories reinforce the importance of our technique for biosurveillance and affirm that synthesis accelerates human-directed gene transfer across the tree of life.

## Results

**Classification scheme for natural or synthetic genes.** Many plausible definitions could be used for defining whether a sequence is natural or synthetic. We define a natural gene sequence as one that is found in naturally occurring genomes or metagenomes, including sequences that contain small deviations such as those resulting from natural evolution or from minor human interventions such as appending of short tags. We also consider complementary DNA (cDNA) sequences as natural given that they can be generated from naturally occurring messenger RNA using reverse transcription. In contrast, we apply the term synthetic to gene sequences that contain significant deviations from any single known contiguous naturally occurring sequence. We determined what constituted a significant deviation empirically by applying machine learning techniques to training

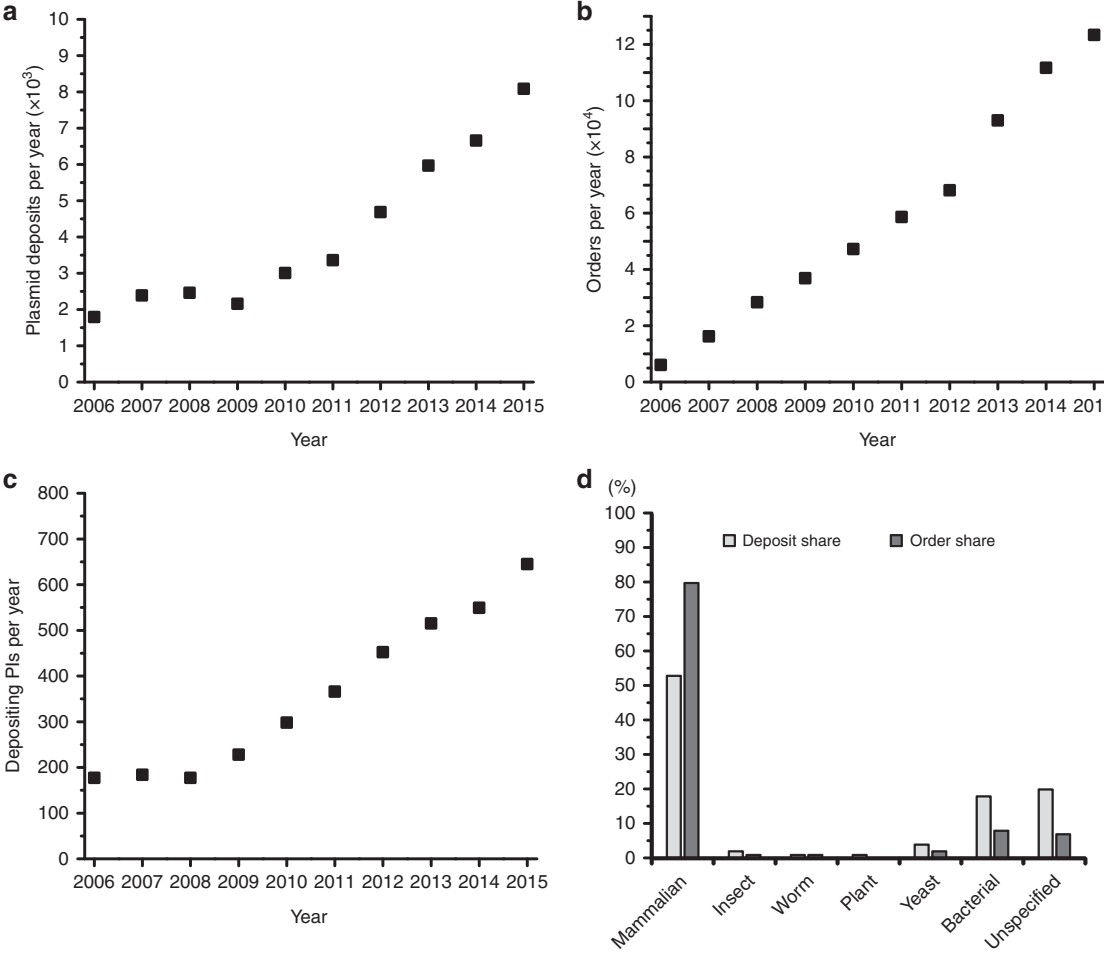

**Fig. 1** Descriptive statistics of the Addgene plasmid repository, a representative source of sequences used in academic biological research. **a** Number of plasmids deposited per year during 2006–2015 in Addgene. **b** Number of orders per year from Addgene. **c** Number of unique depositing principal investigators (PIs) per year. **d** Share of deposits and orders for plasmids across different expression platforms

and test sets of natural and synthetic sequences that we validated manually by sourcing them from sequence databases or from publications. Our definition is pragmatic and has limitations which reflect that we are only using the nucleotide sequence for the classification. For example, it is necessarily the case that if a researcher ordered synthesis of a gene sequence that was identical in every base to a natural sequence, we would classify that gene as natural.

To learn which attributes best predict this classification, we considered two sets of quantitative attributes: intrinsic properties that we could determine from the sequence (such as GC content and rare codon percentage); or comparative properties that we could determine through similarity comparisons with a reference sequence database (such as query coverage—"QCov"—or percentage identity – "%Id") (Fig. 2a, see Methods for full set of properties considered). We hypothesized that most of these properties would improve classification accuracy. To gather the comparative information, we used nucleotide Basic Local Alignment Search Tool[35,36] (BLASTn) to test each sequence against the National Center for Biotechnology Information (NCBI) RefSeq database, a comprehensive database of naturally occurring genomes, metagenomes, and cDNA libraries[37,38], and extracted comparison data for the best alignment entry.

Because many published or publicly disclosed codon-optimization procedures use a weighted Monte Carlo approach proportional to codon abundance (or codon adaptation index, CAI)[39–43], we surmised that there might be an effective cutoff value for %Id below which there should only be synthetic sequences. To quantify this theoretical cutoff, we pursued two strategies. First, we computed the average expected %Id of a nucleotide sequence assuming randomized codon-substitution without weighting by the codon usage of any particular organism. Each codon substitution thus produced an expected %Id based only on the number of codon possibilities for each amino acid and the nucleotide substitutions between them. Weighting these values by the amino acid occurrence frequency in nature[44] indicates that a randomly codon-substituted sequence should, on average, have 78 %Id compared to the starting non-substituted sequence (Supplementary Tables 1–3). This provides the baseline against which we compare our second strategy, which tests the expected %Id from codon-optimization for specific organisms. We did stochastic simulations of all potential pairs of 16 different organisms, using their actual codon usage tables. On average, these simulations provided similar results. For example, expression of human sequences in other organisms (Fig. 2b) had an average of 75 %Id. All simulation averages fell below 85 %Id. Codon optimization across other organism pairs revealed important variation from the 75 %Id average: organisms with highly-dissimilar codon usage produced 65 %Id on average, whereas optimizing organisms with highly-concentrated codon usage back to themselves produced 85 %Id on average (overall distribution summarized in Fig. 2c, the 'shoulders' of which, at 65

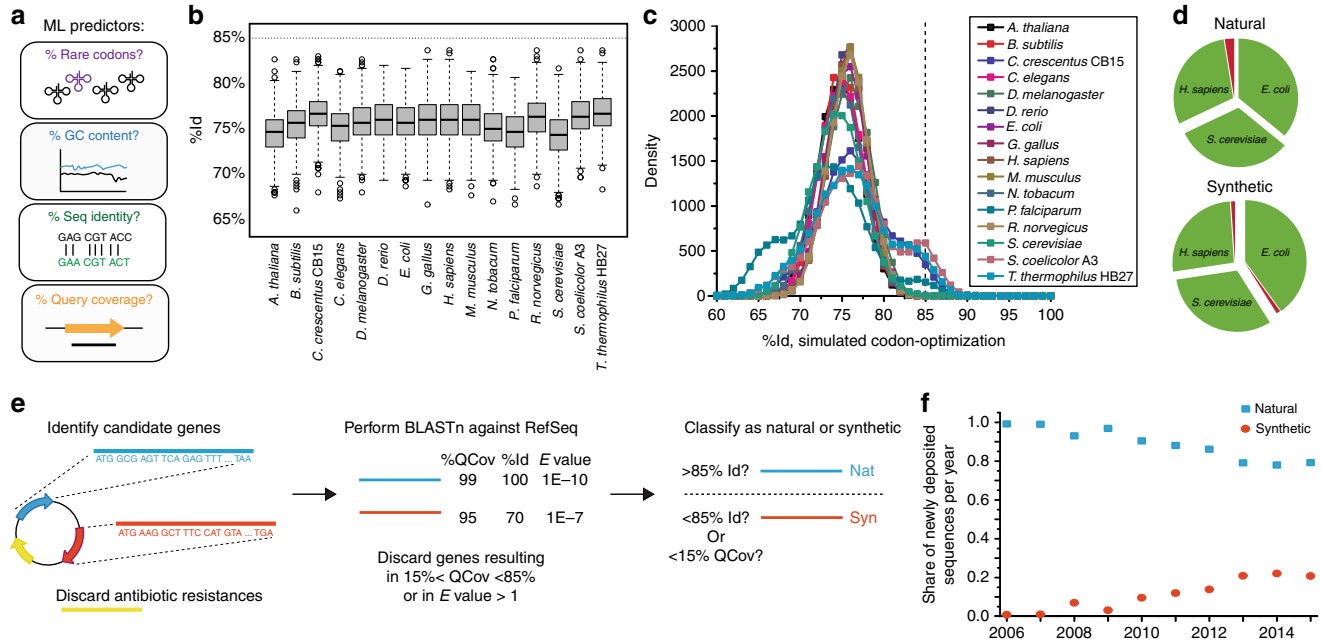

**Fig. 2** A nucleotide-based sequence classification scheme for determining natural and synthetic gene sequences. **a** Cartoon depicting different machine learning input variables that were tested as possible predictors. **b** Simulated sequence percentage identity (%Id) of a codon-optimized sequence originating from 16 organisms (including model organisms and organisms with distinct codon usage) for expression in *H. sapiens*. Shown is the median, surrounded by the gray interquartile range, with whiskers extending to 1.5x the interquartile range, and circles for outliers. **c** Simulated %Id resulting from pair-wise codon-optimization of a hypothetical sequence for all 16 organisms and aggregating by expression organism. **d** Results from application of classification scheme to a manually constructed test set of natural (N = 78) and synthetic (N = 95) gene sequences for expression in *Escherichia coli*, *Saccharomyces cerevisiae* (baker's yeast), and *Homo sapiens*. Regions colored in green represent the proportion of sequences successfully identified by our classification scheme, and regions in red represent unsuccessful cases. **e** Overall classification workflow. **f** Share of natural and synthetic sequences deposited to the Addgene repository, as determined by application of classification scheme

and 85%, represent these extremes). Because model organisms contain more typical codon usage, transfers of genes between two organisms with extreme codon-usage are infrequent. Together our theoretical analysis suggests that Monte-Carlo based codon optimization methods leave telltale signals in the %Id when compared back to the pre-codon-optimization sequence.

To empirically determine which attributes could inform our classification and what quantitative thresholds would be appropriate, we pursued a supervised machine learning approach that considered the full set of previously mentioned variables. We constructed a training set consisting of 83 gene sequences populated with natural and synthetic genes for expression in *E. coli*, *Saccharomyces cerevisiae* (baker's yeast), and *Homo sapiens* (Supplementary Tables 4–11). Synthetic genes included in training and test sets were identified from several independent databases using keyword searches for terms such as "synthetic" or "codon-optimized" and manually verified from user-provided annotation or the methods section from the corresponding publication. We applied random forest machine learning[45,46] to this training set and determined that sequence %Id below 85% was the best predictor of a synthetic sequence, aligning well with our theoretical results. Using this classification criterion on a test set of 173 manually identified sequences yielded 97.7% accuracy (Fig. 2d). This also aligns well with our theoretical simulation results, which predict that 98.6% of synthesized sequences will lie below the 85 %Id threshold. To further validate this threshold, we performed a simple parameter sensitivity analysis using our test set. This demonstrated that other %ID cutoffs are not as effective (Supplementary Table 12). Interestingly, our random forest approach did not identify GC content or rare codon content as an effective predictor.

**Application of the classification scheme to the Addgene data**. We applied our classification scheme to the 19,334 unique genes contained in the Addgene database from 2006–2015 to determine which were synthetic and which were natural (Fig. 2e). For this analysis, we excluded genes encoding known antibiotic resistances based on BLASTn of the Addgene database against reference antibiotic resistance sequences. We also excluded genes that were likely to encode fusion proteins (see Online Methods).

We found that the share of synthetic gene sequences deposited in Addgene has increased over time (Fig. 2f). By 2015, synthetic sequences made up over 20% of the genes in newly deposited plasmids, up from less than 1% in 2006. The increasing abundance of synthetic sequences is consistent with the order of magnitude decrease in the cost of gene synthesis over this period[47,48].

**Examination of differential transgene expression**. Using our classification and BLASTn results, we investigated patterns of source and expression organisms for natural and synthetic gene sequences. Because Addgene expression fields contained terms broader than specific organisms, we grouped expression into six categories: Mammalian, Worm, Insect, Plant, Yeast, and Bacteria (Supplementary Table 13) and use this as the expression organism. We determine the source organism by considering the organism corresponding to the best alignment entry (also known as the maximal-scoring segment pair) for a gene sequence. An alternative approach to finding the source organism would have been to use BLASTx to identify the source organism in addition to BLASTn to identify %QCov and %Id. In practice such an approach has important drawbacks, for example more sparsely populated reference databases (see Online Methods).

Many synthetic sequences resulted in no BLAST alignment to any sequence in the RefSeq database, and these were designated as "No Hit." Sequences that result in "No Hit" are likely to be de novo synthetic sequences that deviate from any known protein. We report the proportion of sequences that fit into this category and where they are expressed because this is of independent interest, but we ignore these sequences for subsequent genetic distance calculations since they lack a source organism. Additionally, for sequences that best aligned to viral sequences, we included a "Virus" category that exists outside of taxonomic relationships for living organisms. We binned all sequences with available source organisms by phylum in accordance with NCBI taxonomic practice.

A precise determination of genetic distance between source and expression organisms is not possible using existing taxonomic systems because they are not quantitative and because such comparisons cannot be made at the phylum level. Instead we estimate genetic distance using 16 S or 18 S ribosomal RNA (rRNA) sequence from the SILVA database[49]. rRNA is highly evolutionarily conserved and can function as an evolutionary chronometer since 18 S rRNA is the eukaryotic nuclear homolog of 16 S rRNA in prokaryotes[50,51]. We constructed a phylogenetic tree using the web tool Phylogeny.fr[52] and extracted genetic distance estimates for each source-expression pair based on the most-common organism in that phylum in the Addgene database (Fig. 3a and Supplementary Fig. 1). These genetic distances represent the fraction of mismatches at aligned positions, as is conventional in phylogenetic analysis. Because we are measuring the distance between the source and expression organisms (and not to the specific query sequence), our measure of genetic distance for the usage of a sequence is independent of whether or not it is classified as synthetic.

We display heatmaps showing the number of natural and synthetic gene sequences in the Addgene database corresponding to source-expression category pairs across the 22 most common phyla (Fig. 3b). From these heatmaps, we can make several observations about the relative magnitude of phylum sourcing, the kinds of gene transfers occurring, and the differences in these aspects between natural and synthetic genes. Though the most common expression system for Addgene plasmids is Mammalian, the largest source of unique gene sequences by a significant margin based on BLASTn is Phylum Proteobacteria. The next largest sources of unique gene sequences are Phylum Chordata, viruses, and Phylum Cnidaria, respectively. This may reflect the relative focus on studying vertebrate and viral genes, as well as the importance of GFP in biological research. Sequences sourced from Proteobacteria are used at approximately similar levels in Mammalian and Bacterial expression systems, regardless of whether the sequence is identified as natural or synthetic. On the other hand, sequences sourced from Chordata are predominantly used in Mammalian expression systems, regardless of whether the sequence is identified as natural or synthetic.

These heatmaps demonstrate significant transgene expression for both natural and synthetic gene sequences. The most frequent type of transfer is from source phylum Proteobacteria to Mammalian expression. Though this may be consistent with the predominance of deposits and orders of mammalian expression constructs from Addgene, it is striking that the frequency of transfer from the source phylum Chordata to bacterial expression (essentially the reverse phenomenon) is far lower. A higher-level pattern observable in the heatmaps of Fig. 3b is their relationship with genetic distance shown in Fig. 3a. If genes were being most commonly expressed in their source organism, one would observe hotspots in Fig. 3b along a diagonal axis roughly from upper-left to lower-right. These hotspots are clear for animal expression platforms for both natural and synthetic genes. For natural genes, the pattern extends into many bacterial sequences (hotspots on the lower-right). However, for synthetic genes there is a marked change in the trend for bacterially sourced sequences. Hotspots frequently appear on the lower-left, indicating a high-frequency of mammalian expression of bacterially derived, synthetic sequences.

**Genetic distances between source and expression organisms**. From these heatmaps it is difficult to quantify the differences in expression of natural and synthetic genes. Thus, we calculated genetic distances between the source and expression organism for each sequence. Overall, consistent with our main hypothesis, we find that the average genetic distance between source and expression organisms is greater for synthetic than for natural gene sequences, and that this distinction is highly statistically significant. Table 1 shows these results through a series of regression specifications. In all cases, Synthetic is a binary variable which is 1 if our classification system deems that sequence synthetic, and 0 otherwise. Specification (1) shows that expression with synthetic sequences is, on average, 0.077 units (t-test p-value < 0.01) farther from the source organism than are natural sequences. Specification (2) shows that the gap between the genetic distance between synthetic and natural sequence use grows with sequence length, with each extra kilobase adding 0.117 units (t-test p-value < 0.01) to the difference. Specifications (3) and (4) confirm the finding of specification (2), but use the alternative dependent variable Cross Kingdom, which is a binary variable equal to 1 if the expression is cross-kingdom and 0 otherwise. These trends remain even if CRISPR-Cas9 sequences are excluded from the analysis (Supplementary Table 14). Figure 4 uses a non-parametric local regression (loess) to show the relationship between gene length and genetic distance, for both natural and synthetic sequences. The shaded regions represent one standard error.

Our results suggest that the longer a natural gene sequence is, the less likely it is to be transferred into another organism by researchers. This observation is consistent with the perception that longer unmodified gene sequences are generally more difficult to express and that, as sequence length grows, so does the likelihood that there will be sequence regions that are troublesome to express in another organism. In contrast, synthetic sequences experience little to no drop in genetic distance as gene length grows, and at large lengths are used predominantly for transfer across distant organisms. Thus we conclude that gene synthesis enables transgene expression at a much higher rate than traditional techniques, and that this difference is both scientifically and statistically significant.

## Discussion

This paper introduces a nucleotide-only-based method for determining whether a genetic sequence is synthetic or natural. Grounded both in codon theory and in empirical testing using machine learning, we find that we can correctly predict with 97.7% accuracy on a novel data set. The key heuristics that enable this classification are the percentage nucleotide identity and query coverage of a gene sequence compared across a reference database of natural sequences. Somewhat surprisingly, our machine learning approach did not find GC content or rare codon usage to be an effective predictor. Very usefully, BLASTn queries against the RefSeq genomic collection simultaneously provide the data needed for sequence classification, as well as the organismal origin of the gene.

Using our classification method and phylogenetic distance calculations on sequences in the Addgene database, we provide empirical evidence that gene synthesis is being widely used by practitioners to source genes from genetically-distant organisms,

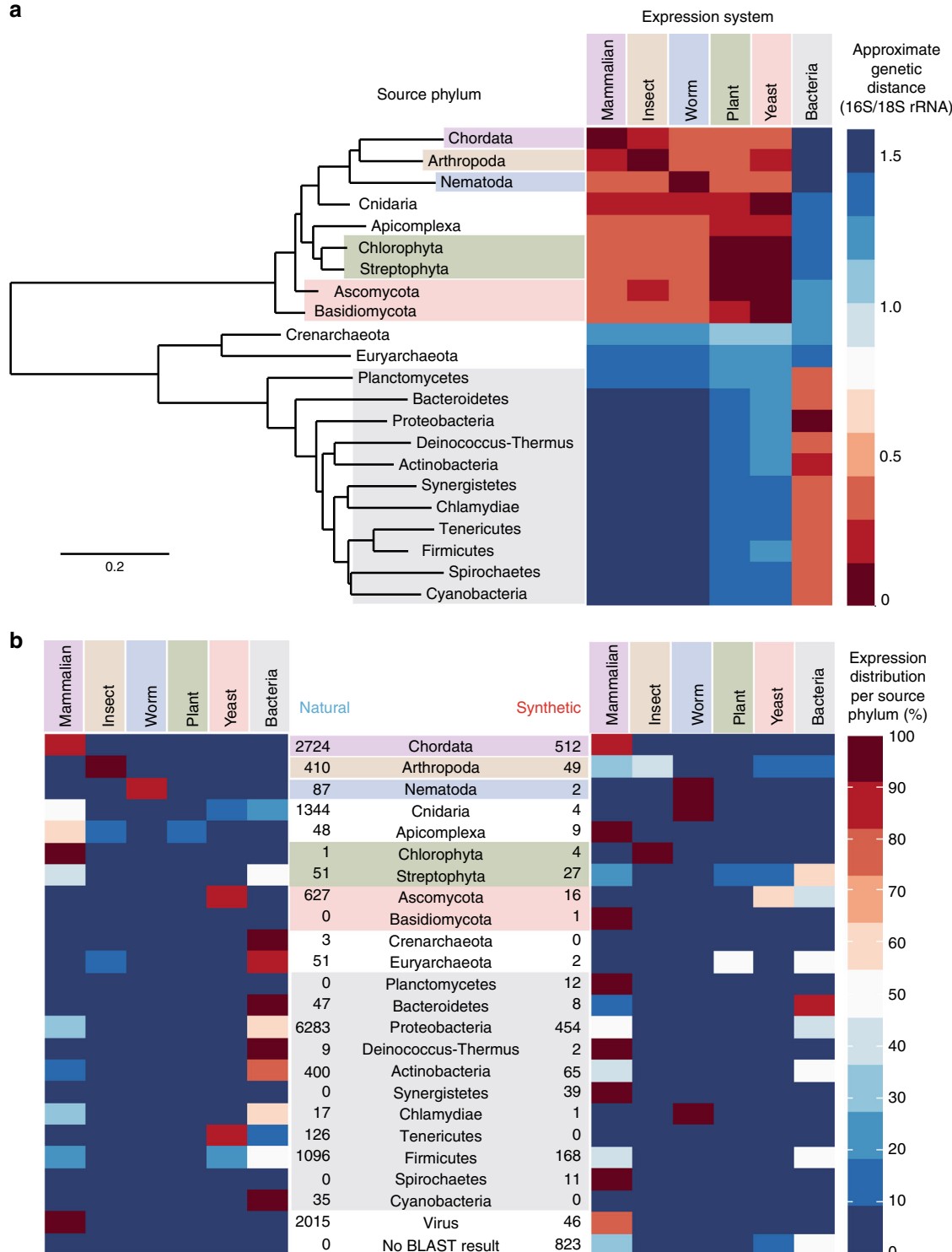

**Fig. 3** Mapping of genetic distances and transgene expression for Addgene sequences. **a** Phylogenetic tree of the most common source phyla and corresponding heatmap displaying genetic distance of different expression platforms. **b** Heatmaps displaying source phylum and expression platform for (left) natural and (right) synthetic genes

which is a finding of important consequence for biotechnology and biosurveillance communities. The genetic distance between the organism used for gene expression and the organism from which the gene was sourced is not only notably more distant for synthetic rather than natural sequences, but this gap grows as sequence length increases. Our finding sheds light on the tension in using synthesis for longer gene sequences. On one hand, a longer natural gene sequence would be more likely to contain codons problematic for gene transfer, making synthesis more attractive for these sequences. On the other hand, methods and pricing for synthesis vary widely based on DNA length[15], and thus community behavior may be influenced by many factors including size limits on common synthetic gene offerings (e.g., gBlocks from Integrated DNA Technologies) towards not using

**Table 1 Regression results comparing natural and synthetic sequences from the Addgene Database**

| | Dependent variable: | | | |
|---|---|---|---|---|
| | Genetic distance | Genetic distance | Cross kingdom | Cross kingdom |
| | OLS | OLS | OLS | Logit |
| | (1) | (2) | (3) | (4) |
| Constant | 0.499*** (0.006) | 0.606*** (0.009) | 0.572*** (0.007) | 0.452*** (0.035) |
| Synthetic | 0.077*** (0.018) | −0.041 (0.029) | −0.017 (0.022) | −0.228** (0.093) |
| Gene length [kb] | | −0.112*** (0.007) | −0.098*** (0.005) | −0.587*** (0.035) |
| Gene length [kb] * Synthetic | | 0.117*** (0.013) | 0.076*** (0.010) | 0.495*** (0.049) |
| Observations | 14,745 | 14,745 | 14,745 | 14,745 |
| $R^2$ | 0.001 | 0.018 | 0.023 | |
| Adjusted $R^2$ | 0.001 | 0.018 | 0.023 | |
| Log likelihood | | | | −10,000 |
| F statistic | 17.82 | 91.8 | 115.72 | |

*$p < 0.1$; **$p < 0.05$; ***$p < 0.01$

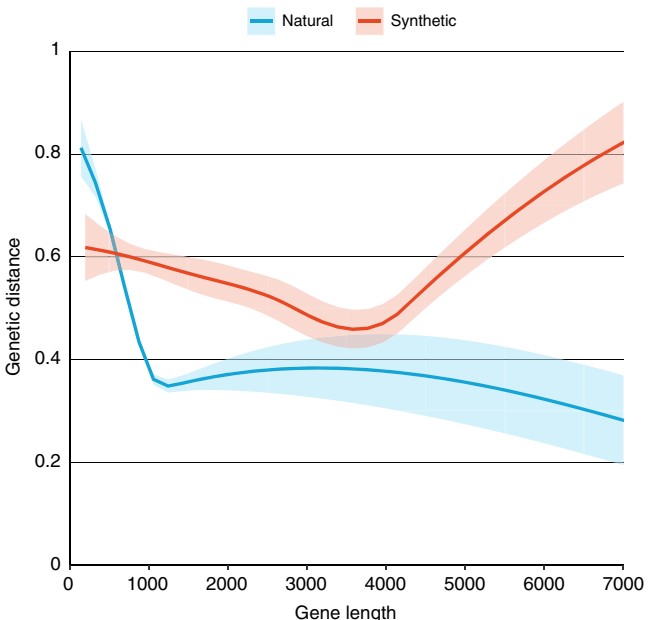

**Fig. 4** Local regressions (LOESS) of genetic distance and gene length for natural and synthetic sequences in the Addgene database. Bands show one standard error in each direction, loess smoothed (span = 0.9)

synthesis for long sequences. Our results suggest that, at the margin, scientists are more influenced by the ability of gene synthesis to access the treasure trove of natural genetic diversity and transfer it to new organisms.

Determining the provenance of a genetic sequence is typically the first step of forensic attribution associated with bio-surveillance. As gene synthesis technology is further democratized and genetically-engineered organisms increase in capability, such sequence classification tools are vital to identifying and monitoring engineered organisms that may be accidentally or deliberately released into new environments. Commercial gene synthesis suppliers already provide some security in this area by screening orders for potentially hazardous sequences[24]. But, as a major U.S. bipartisan biodefense study[27] and ongoing U.S. intelligence agency research programs[28] recently highlighted, there are limited tools to detect engineered organisms, which may have been constructed by circumventing gene synthesis regulations. In these circumstances, there is significant value in being able to analyze the sequences after-the-fact, for example based on an environmental sample obtained from a suspicious site.

The classification method reported here can form part of a suite of tools and strategies that help identify an engineered organism (see Supplementary Discussion for a proposed workflow). Once an organism of interest is isolated, conventional tools can be used for whole genome sequencing, de novo genome assembly or reference genome alignment, and then open reading frame (ORF) detection. Our classification scheme can subsequently be applied to a subset of ORFs or every ORF to identify synthetic genes. Since transfer of natural genes is also of interest for biosecurity purposes, our more general approach of using existing BLASTn and phylogenetic tools to examine ORFs can help identify transgenes and evaluate the likelihood of horizontal or engineered transfer. If an engineered organism cannot be isolated and is part of an impure environmental sample, additional approaches such as 16 s rDNA sequencing and knowledge of environmental baselines may be needed. Upon identification of a synthetic gene, BLAST can provide functional annotation and guide response strategies to the engineered organism harboring the synthetic gene. For example, these approaches would

accurately identify a recently engineered yeast strain designed to produce opioids as numerous synthetic genes were required to achieve this feat[53]. Other approaches would be needed to identify engineering modifications to non-ORF regions as these are outside the scope of our tools.

A particularly apt setting for our approach is agricultural risk. For decades, through the Coordinated Framework for Regulation of Biotechnology, the USDA Animal and Plant Health Inspection Service has had oversight of genetically engineered organisms that may pose agricultural risk[29]. However, genetically modified organism (GMO) detection in agriculture has been limited to PCR-based methods with primers designed to target known genes associated with GMOs, which are most commonly synthetic transgenes[54,55]. Detection of GMO crops or food ingredients is of heightened interest in the European Union given stricter regulation. While state of the art methods for GMO detection in the EU have featured more extensive databases, they remain associated with PCR methodology. The Joint Research Center (JRC) of the European Commission constructed the GMOMETHODS database which contained 118 different PCR methods allowing identification of 51 single GM events and 18 taxon-specific genes in a sample as of 2012[56]. In 2015, the JRC followed up with a database specifically aimed at storing GMO-related sequences called JRC GMO-Amplicons[57]. Though the database name refers to amplicons, the authors note that the availability of an updated GMO sequence databases has increased relevance after the advent of next-generation sequencing. When coupled with next-generation sequencing, our classification method should provide a more general and complementary approach to comparisons against known GMO-associated sequences because it can identify uncatalogued synthetic transgenes, such as those not intended for crop enhancement. In addition to agriculturally oriented agencies, public health, environmental, and biosecurity agencies would benefit from the ability to screen for untargeted genes in organisms to identify unusual risks. Engineered organisms containing synthetic genes are of particular interest because in academic settings they have been demonstrated to produce non-native illicit substances[53], to express non-native toxins[58], or to execute

complex programs designed to alter human cell fates[59]. Furthermore, engineered microbes released into the environment can persist for years[60,61]. Because the RefSeq reference genome collection includes pathogen genomes, our classification approach can also be used to identify gene transfers into known pathogens.

As an enabling technology for life science research, gene synthesis has changed the behavior of scientists. In the absence of affordable gene synthesis, researchers could look for parts in a narrow genetic neighborhood where transfer would be relatively easy, or source more broadly across organisms at the cost of potentially incurring much greater engineering effort with little guarantees of eventual success. Today those same researchers can source their parts from wherever makes the most biological sense with the knowledge that gene synthesis should help them overcome one of the main expression challenges. As such, gene synthesis could allow biologists to source genes from farther away in the tree of life. This paper shows that it does, providing a bird's-eye view of the community's preference for relying on gene synthesis to transfer genes across large genetic distances. This trend promises to be of scientific, industrial, and medical use, to the great benefit of biologists and society at large. At the same time, society must be prepared in the event of accidental or deliberate release of genetically engineered organisms, and tools for synthetic sequence identification constitute a foundational part of these efforts.

## Methods

**Codon-substitution sequence percentage identity calculation**. As a first approximation for a cutoff value for sequence percentage identity, we calculated the expected sequence identity of any given gene sequence after codon substitution, without accounting for the relative differences in codon usage (Supplementary Table 1). First, we determined the expected percentage identity associated with codon-substitution at the amino acid level. Barring very rare exceptions, the 20 canonical amino acids are each encoded by the same codons throughout all known life. For all but three amino acids, the third nucleotide in the codon is the only variable position. At the codon level, this means that the sequence encoding an amino acid with two codon choices will either remain identical after optimization (3/3 bases unchanged) or be 67% identical (2/3 bases unchanged). Thus, for amino acids with only two synonymous codon choices, the average expected sequence percentage identity is 83%. Similarly, amino acids with four codon choices have an average expected percentage identity of 75% after codon-substitution. The three amino acids feature nucleotide changes at positions other than just the third position. These each have six codon choices. For two of these amino acids—R (arginine) and L (leucine)—there are two codons where the first nucleotide also varies. This case is illustrated in Supplementary Table 2 for R. An R or L codon is expected to have 61% sequence identity on average after optimization. In the case of S (serine) (Supplementary Table 3), two of the six codon choices have differences in the first and second nucleotide in addition to the usual third nucleotide variation. Thus, after determining the expected percentage identity associated with codon-substitution for each amino acid, we obtained a weighted average of 78% using the natural frequency of occurrence of each amino acid[44]. Creating a threshold level that separates natural and synthetic would need to be higher than this, to account for random variance of codon usage. To do this, and more precisely align with actual amino acid usage, we performed a simulation.

**Codon-optimization stochastic simulation**. We performed a stochastic simulation to model the transfer of genes between 16 organisms: *A. thaliana*, *B. subtilis*, *C. crescentus* CB15, *C. elegans*, *D. melanogaster*, *D. rerio*, *E. coli*, *G. gallus*, *H. sapiens*, *M. musculus*, *N. tobacum*, *P. falciparum*, *R. norvegicus*, *S. cerevisiae*, *S. coelicolor* A3, and *T. thermophilus* HB27. Codon usage tables for these organisms were obtained from the Codon Usage Database (http://www.kazusa.or.jp/codon/). We considered every pairwise transfer of genes within these (including transfer back to the organism itself) and modeled what percentage identity would be expected upon codon optimization. In short, we completed the following steps: (i) Source organism amino acid sequence—Created a random sequence of 1000 amino acids, based on the natural occurrence rate of such amino acids in the source organism; (ii) Source organism nucleotide sequence—For each amino acid in the source organism amino acid sequence we randomly chose a codon that represents it, weighting the choice by the source organism's codon usage table; (iii) Expression organism nucleotide sequence—For each amino acid in the source organism amino acid sequence we randomly chose a codon that represents it, weighting the choice by the expression organism's codon usage table; (iv) Comparison of the Source Organism and Expression Organism nucleotide sequences—we compare sequences codon-by-codon to determine whether they are identical. The set of steps was

repeated 1000 times for each of the $16^2$ pairings, yielding 256,000 simulation runs. R code used for the simulation can be found in the Supplemental Code section.

**Determination of classification criteria**. To identify a suitable classification criteria, we compiled a set of variables that could potentially determine whether a part is naturally occurring ("natural") or was produced synthetically ("synthetic"). Initially, we considered the following variables: Percentage of rare codons (less than 2% occurrence in the host); Percentage of rare codons (less than 5% occurrence in the host); Percentage of rare codons (less than 10% occurrence in the host); Average codon abundancy; GC content; BLAST output variables, such as maximum query coverage, maximum percent identity, maximum percent identity with query coverage greater 50%, maximum percent identity with query coverage greater 50%, maximum percent identity with query coverage greater 85%, maximum percent identity with query coverage greater 95%, number of hits with query coverage greater 50%, number of hits with query coverage greater 85%, and number of hits with query coverage greater 95%. We also tested different combinations of the above variables to assess potential multiplicative or correlative effects.

Host codon occurrences were determined from OpenWetWare for *E. coli* (http://openwetware.org/wiki/Escherichia_coli/Codon_usage) and the Kasuza Codon Usage Database for *S. cerevisiae* and *H. sapiens* (http://www.kazusa.or.jp/codon/).

**Construction of training and test sets for empirical testing**. To gain a sense of the percentage sequence identity differences that we would observe and to test the influence of other variables, we constructed a training set consisting of synthetic sequences that were known to be codon optimized for expression in specific organisms and a control set of natural sequences. A complete description of the training and test sets is included in Supplementary Methods.

**Procedure for variable reduction using random forest**. To determine the most useful set of variables that would distinguish between natural and synthetic sequences, we applied the R package random forest ('randomForest'—https://cran.r-project.org/web/packages/randomForest/randomForest.pdf). Random forest is a learning method that can be used for classification by constructing a multitude of decision trees using a training data set. With a test set, the individual trees output the mode of the classes. We observe that percent identity is sufficient to predict whether a sequence occurs naturally or was made synthetically. Additional variables did not improve the classification result. Using our training set of sequences, we identified 85% identity as the threshold. Sequences that have a higher percent identity when performing BLASTn against the RefSeq database can be classified as natural, while sequences with a lower percent identity are likely produced synthetically.

**Approach for nucleotide BLAST (BLASTn)**. To align sequences pairwise or to a database, we used the NCBI BLAST + suite. We calculated pairwise alignments using the standalone version on a local machine (ftp://ftp.ncbi.nlm.nih.gov/blast/executables/blast+/LATEST/—version 2.5.0). For alignments to a larger data base such as RefSeq (see below), we used NCBI BLAST + on Amazon Web Services (https://aws.amazon.com/marketplace/pp/B00N44P7L6/ref=mkt_wir_ncbi_blast#—version2.5.0). In both cases, since we only have nucleotide sequences in our database and are looking for only related sequences, we use the 'BLASTn' algorithm and apply following parameters: Maximum target sequences = 999,999; Expect threshold = 1; Word size = 11 (4 for pairwise alignment); Match score = 2; Mismatch score − = −3; Gap cost − Existence = 5, Extension = 2.

We chose to perform BLASTn against the Reference Sequence (RefSeq) database. RefSeq is maintained and provided freely by the National Center for Biotechnology Information (NCBI) and is, to our knowledge, the most comprehensive database of the genetic sequences found in natural organisms[37,38].

**Application of classification scheme to Addgene data**. For the alignment of all Addgene sequences against the RefSeq data base, we used NCBI BLAST + on Amazon Web Services. We ran a c3.8xlarge instance (https://aws.amazon.com/ec2/instance-types/?nc1=h_ls) with 32 virtual CPUs and 60 GiB memory. The BLAST + suite contains only the tax ID for each entry. To access the kingdom and scientific name of each hit we use the taxonomy database (ftp://ftp.ncbi.nlm.nih.gov/blast/db/taxdb.tar.gz). We ran the BLAST+ commands directly on the Amazon Web Service instance using this command line option:

blastn -query AddgeneSequenes.fasta -db refseq_genomic -evalue 1 -max_target_seqs 999999 -word_size 11 -gapopen 5 -gapextend 2 -penalty −3 -reward 2 -outfmt "6 qseqid sseqid sacc sskingdoms staxids sscinames scomnames length evalue pident nident mismatch qcovs qcovhsp qstart qend sstart send" -out RefSeq_AddgeneSequenes.txt.

We are grateful to Addgene for sharing their data with us for this research project. The data was received in multiple CSV files. The Addgene data contains a wide range of information for each plasmid in the repository. For this research project we focused on the following information: Plasmid name/ID; The year a plasmid was deposited with Addgene; Plasmid expression system (vector type); Plasmid sequence; Features on the plasmid (e.g., ORFs, ribozyme binding sites,

promotors) and the start and end position of each feature. We subsample the available data and only considered plasmids for which a submission date, a full sequence, and a list of annotated biological parts was provided.

Two pieces of information from the previous list needed to be cleaned for this research project. The 'features' information was cleaned and summarized to reduce the computational power that was needed to align the sequences to the RefSeq data base. We removed all features except for ORFs. Theoretical ORFs within plasmid sequences were detected by Addgene. Prior to June 26, 2017 (the launch date of SnapGene-powered maps), Addgene in-house software was used to detect theoretical ORFs. An arbitrary minimum ORF length of 150 amino acids was set and start codons (ATG) were searched for in all six reading frames. We then aligned all Addgene-identified ORFs pairwise using BLAST+. Each sequence received a unique ID, and if two sequences had 100% query coverage and 100% identity, then the same ID was given to the identical sequences.

We also cleaned the plasmid expression system information by converting each entry into one of seven simplified expression categories. This was necessary because the information is not curated by Addgene and scientists often add more than one expression system. This was done by making two assumptions: (1) that information with multiple entries were most likely cloned in a lower life form and primarily expressed in the highest life form listed; (2) that information containing viral expression platforms were primarily intended for mammalian expression. Supplementary Table 13 lists the full set of original categories and the corresponding simplified expression category assigned.

**Exclusion of antibiotic resistance gene sequences**. Antibiotic resistances are used differently than other genes. Notably, the requirement of every plasmid to contain an antibiotic resistance or other selective marker means that a small number of genes are used highly redundantly. In addition, antibiotic resistances have been acquired by natural pathogens of high medical interest and therefore synthetic versions of these sequences are more likely to be found in the RefSeq database, potentially leading to false natural classification. Therefore, we removed them from our sample. First, we retrieved the most common antibiotic resistances from the Addgene website (http://blog.addgene.org/plasmids-101-everything-you-need-to-know-about-antibiotic-resistance-genes) and created a list of all the features in the Addgene database that are labeled as one of the antibiotic resistances. We retrieved the sequences of these features and built a database that we aligned to all other sequences from the Addgene database. If an ORF shares more than 85% query coverage and more than 85% identity with one of the previous identifies antibiotic resistances, then we consider it as an antibiotic resistance. With this approach we identified 534 unique antibiotic resistance sequences in our sample, and we excluded these sequences from further analyses.

**Exclusion of sequences with query coverage between 15–85%**. As described above, in our test and training sets we classified sequences that with a more than 85% identity hit in the RefSeq data base as natural. In the Addgene data we add one additional constraint, reflecting the usage of fusion proteins (which were not in our training or test data). This is important because we would not want to classify an entire sequence as natural if 50% of the sequence has 100% sequence identity, whereas the other 50% has 0%—but this is exactly the result that would be obtained if we ignore the query coverage (which reveals this percentage of the sequence that is being matched).

We impose an additional requirement that genes must also have more than 85% query coverage to be deemed natural. Sequences that have less than 15% query coverage with any sequence in the RefSeq database, or those that result in "No Hit", are likely to be so unnatural (true de novo sequences) as so to be considered synthetic. As already mentioned, in our training data we observed too few instances with low query coverage and high percentage identity, to determine a precise tradeoff for how much query coverage would be optimal. Nevertheless, we chose an 85% query coverage cutoff as a form of robustness against misclassifications with BLASTn. This was guided by not wanting to pick too high a level, lest we exclude natural sequences with added tags for common purposes such as purification or localization, which are often 20–100 base pairs (and therefore less than 10% of a standard 1000 base pair gene). Similarly, we did not want to pick too low a level, since BLASTn searches preferentially for highly identical regions, and thus might cut off the end of a synthetic sequence yielding a maximal-scoring segment pair with lower query coverage and higher percentage identity (thus falsely classified as natural).

For sequences with less than 15% query coverage, we assume that they are fully synthetic. This threshold is somewhat arbitrarily to reflect that longer sequences are unlikely to be fusion proteins (and to be consistent with an upper threshold of 85%).

We hypothesized that sequences resulting in query coverages between 15 and 85% are very likely to be fusion proteins. We test this for the set of putative fusion proteins by removing the portion of the query that successfully aligned in the first BLAST and re-running BLAST on the remaining shortened sequence. We found that many of the remaining sequence queries aligned with high query coverage on the second BLAST, suggesting that they were indeed fusion proteins.

**Regression analysis**. For the regression analyses shown in Table 1 we used the software package StataIC 12(details in Supplementary Code). We ran ordinary least square regressions (OLS) for the following specifications:

1.  **Genetic Distance** $= \alpha + \beta$ **Synthetic**
2.  **Genetic Distance** $= \alpha + \beta$ **Synthetic** $+ \gamma$ **Gene Length** $+$ $\psi$ **Gene Length** $\times$ **Synthetic**
3.  **Cross Kingdom** $= \alpha + \beta$ **Synthetic** $+ \gamma$ **Gene Length** $+$ $\psi$ **Gene Length** $\times$ **Synthetic**

In specification (4) we repeated the third specification but estimated it using a logit regression, to reflect the binary outcome variable.

In Fig. 4 we estimate two regressions, one on synthetic genes and the other on natural one, using the loess regression function in ggplot (details in Supplemental Code). In each case we used a local regression of GeneticDistance on GeneLength with a span of 0.9.

**Evaluation of transgene expression**. We assigned a source organism for each gene sequence based on the organism of best BLASTn alignment. To determine source organisms, we faced a choice of whether to use BLASTn or BLASTx (which translates the queried nucleotide sequence into a protein sequence and searches for that). We chose to use BLASTn for several reasons. First, the use of BLASTx would require an additional BLAST run. Second, although protein-based BLAST strategies are recommended for determining the structure and function of proteins encoded by genes, BLASTx may be less accurate than BLASTn for source organism determination because NCBI protein collections are more sparsely populated than NCBI nucleotide collections and because protein sequences are more highly conserved. Spot testing confirmed this, with BLASTx appearing to offer lower resolution than BLASTn for source organism identification. In future studies, one could envision evaluating a wide range of BLAST strategies for source organism determination, including weighting nucleotides by codon position. In any case, we expect that these differences in source organism assignment would be negligible if organisms were grouped by phylum as we have done. To compare the performance of BLASTn and BLASTx in determination of the phyla of source organisms, we manually evaluated 50 randomly chosen sequences from the Addgene data. In 25 out of the 50 sequences, the two approaches led to differing conclusions for the maximal-scoring segment pair (most of which had lower %QCov and %Id scores from BLASTx than from BLASTn). Within these, however, only 5 displayed differences in phylum.

We determined the expression category using the cleaned Addgene data for expression system (Supplementary Table 13). We determined a representative organism for each phylum by most common member of that phylum in the Addgene database and obtained 16 S/18 S ribosomal RNA sequences for each organism (see Supplementary Fig. 1).

**Code availability**. All the authors' analysis code is included in the Supplementary Information.

Our analyses were run on: R version 3.3.1 (for the codon simulation) and 3.4.1, Python version 3.5, and Stata version IC12.

## Data availability

The following data were used for this paper: Addgene plasmid data: proprietary to Addgene, viewable but not downloadable at https://www.addgene.org/. RefSeq reference genome collection: Available at https://www.ncbi.nlm.nih.gov/refseq/. Codon Usage Databases: Available at https://www.kazusa.or.jp/codon, and https://openwetware.org/wiki/Escherichia_coli/Codon_usage. SILVA 16S rRNA Database: Available at https://www.arb-silva.de/. All other data available upon request from the authors. Machine readable versions of the data presented in the Supplementary Information are available at https://github.com/AKunjapur/Synthetic-gene-classification.

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

## Acknowledgements

We are tremendously indebted to Addgene for sharing their data, answering questions as they arose, and providing manuscript feedback. We thank Dr. Alec Nielsen (MIT), Dr. Darrell Ricke (MIT), and Dr. James Comolli (MIT) for discussions about BLAST. We are grateful to Dr. Nili Ostrov (Harvard) and George Chao (Harvard) for discussions about genetic distance calculations. A.M.K. was supported by a National Science Foundation Graduate Research Fellowship. P.P. was supported by two grants of the European Union's Seventh Framework Programme FP7. The collaborative research project ST-FLOW (KBBE-2011-5—Grant Agreement number 289326) and the People Programme (Marie Skłodowska-Curie Actions—Grant Agreement number 612614) and N.C.T. was supported by a grant from MIT.

## Author contributions

A.M.K. and N.C.T. conceived the study. P.P. tabulated Addgene descriptive statistics and developed/implemented classification scheme under guidance from A.M.K. and N.C.T. A.M.K. and N.C.T. performed codon-substitution and codon-optimization analyses, respectively. P.P. performed all data analysis of BLAST results, genetic distances, and regressions under guidance from N.C.T. A.M.K. performed phylogenetic analysis to determine genetic distances and categorized source/expression organisms. A.M.K. and N.

C.T. jointly led manuscript writing and A.M.K. led figure production. All authors read and approved the final manuscript.

## Additional information

**Competing interests:** The authors declare no competing interests.

