## [Peer Review File · Nature Communications]

Reviewers' Comments:

Reviewer #1:

Remarks to the Author:

Research presented in this paper works toward the development of methods to detect purposefully engineered organisms (from natural organisms) in the context of biodefense. This is an important need for microbial forensics, and the reason for such programs as FELIX; it is also extremely challenging and the authors have devised a novel means for attempting to do this. The methods here are heavily caveated – and rely on a number of built in assumptions. (For example, use of a dataset which does not likely represent the sampling environment in the context of a bioevent). That said, the ideas here regarding measurable genetic distance (and codon optimization) of synthesized genes vs natural genes, are compelling tools which could be applied to the detection challenge. My questions in the biosecurity context relate to the applicability of the tools operationally, as considerations for the authors.

The authors reference the production of opioids in yeast, or the production of toxins in *E. coli*, as examples for biodefense concern, which they are – however these particular examples are not ones for which their analysis would provide the most insight in terms of forensics. Actors who use synthetic biological means to produce illicit drugs or toxins at scale would use the compounds themselves for harm, rather than the microbes that created them (one reason that biological creation of chemicals is so concerning is that many actors have long learned viable means for delivery of chemical toxins, as opposed to the much more challenging delivery of biological pathogens, for example). That said, the discovery of such microbes could possibly occur in forensic investigation, if access to a suspected actor's production facility was available for sampling. The methods being developed by the authors could best be applied to contamination of food, water, or the environment by engineered microbes. In that context the authors may want to speculate on how their methods could be useful when the forensic sample would necessarily be less pure: coming from soil, mixed aqueous materials, or animal or plant tissues (this is alluded to at the top of page 10 but may warrant further exposition). Similarly the authors may want to note if environmental baselines (assessments of soil, water, etc. for standard organisms and their nucleotide identities) would be useful.

The authors have left pathogens out of their study, citing "they are already on watch lists for biosurveillance". However, pathogens are of high primary interest to microbial forensics – how can authorities discern a naturally occurring outbreak from one that was initiated through an engineered pathogen? Biosurveillance serves a public health requirement – it does not tell us if we are under a biological attack, and the list of diseases for biosurveillance may not cover novel organisms. Authorities would be unable to attribute such an attack without means to identify that the causative agent had been engineered. It would be helpful for the authors to think about this in the context of developing their tool. For example, would their tool be able to discern if an antibiotic resistance gene were engineered into a common microbe? Or if a fungal species was engineered for temperature resistance and could thus infect humans? That would have immense utility for forensics (similarly, other genes such as those that might confer pathogenicity or other traits in microbes).

I concur that advances in governance of gene synthesis technology can be extremely useful in biosecurity mitigation. I would also note that tools to limit undesired proliferation of engineered organisms is extremely important for biosafety; they would not help, however, in the security context, as they would not be utilized by bad actors.

Other comments

The authors note that the advent of gene synthesis tools are expanding the number and complexity of genes that have been deposited in Addgene. This also likely represents a smaller fraction of what might be more widely occurring in small start ups that are bioprospecting for genes that expressed high value compounds (re sequencing of fungal genomes, soil microbe sampling and sequencing). Whether these genes/gene systems will make their way into shared databases remain to be seen, as some may become proprietary over time.

While plants seem to be in small quantity in Addgene per fig 1D, the authors may want to revisit

some of the literature on genomic detection of genetically modified foods/crops/seeds to determine if their tool has potentially improved utility in contrast to methods currently used in the agriculture realm.

Small typo:

Top of page 4, I believe the word "repositories" should be 'repository".

I hope these comments are useful – thank you for the opportunity to provide a biosecurity review.

Reviewer #2:

Remarks to the Author:

In manuscript NCOMMS-18-01285A-Z, Thompson et al. describe an algorithm to identify synthetic DNA sequences from natural gene sequences. In the second part of the manuscript, they apply this algorithm to sequences extracted from Addgene database. This analysis allows them to identify major trends in the way gene synthesis is used by the life science community.

The results of each section are interesting and timely. The criteria used to identify synthetic genes are sophisticated and appear to be very effective. The conclusions of the Addgene analysis are interesting if not completely unexpected.

Major Revisions

1- The focus of this paper seems to be shifting between the introduction and the conclusion. The introduction is mostly focused on bio-surveillance and the identification of synthetically engineered organisms. The conclusion is mostly focused on analyzing trends in the use of gene synthesis. In my opinion, these are two different questions and I am not sure it makes sense to bundle them in the same manuscript.

2- The classification of genes in synthetics vs. not synthetic is fine for analyzing the Addgene repository but it is very underdeveloped to claim it would be useful in bio-surveillance. There is no critical analysis of its limitations or its benefits compared to other methods. For instance, the algorithm proposed by the government to screen synthetic DNA orders (doi:10.1038/nbt.1802) would be able to detect that different segments of a large DNA sequence are associated with different species. I am not convinced that the codon optimization criteria proposed by the author would result in a major gain of performance. In order to claim that this approach can be used in security applications, I think that the performance of the algorithm would need to be characterized much more thoroughly. It would need to go beyond simply detecting that a gene has been synthesized commercially.

3- In order to address 1 and 2, I would suggest rewriting the introduction to deemphasize the bio-surveillance aspect and simply introduce the classification algorithm in the context of the Addgene analysis.

4- The first paragraph of page 5 is not very clear. My understanding from the figure is that analysis is performed at the gene level but this is not stated anywhere. Also, it is not clear if the analysis is performed based on sequence annotations or for all open reading frames. Finally, I don't understand why the authors use a nucleotide BLAST search to identify the source organisms. It seems to me that they would get better results by doing a BLAST on the translated DNA sequence as protein sequences would be unaffected by codon optimization.

5- The data used in this report should be included in the online supplement or deposited in a public repository such as Figshare. The tables S4 and onward are not useful in their current form. I would like to see the sequences of these genes made available in a computer-readable format. It would also be very interesting to have access to the Addgene primary data. The data should be accompanied by an open data license.

Minor Revisions

1- I think that citing a review about gene synthesis would be appropriate in the second paragraph of the introduction. I suggest 10.1016/j.tibtech.2008.10.007 and doi:10.1038/nmeth.2918

2- The supplement includes the scripts used in this work. They would be more useable if provided

separate text files. I would also encourage the inclusion of an OSI-approved open source license file in the directory containing the scripts.

Jean Peccoud
jean.peccoud@colostate.edu

Reviewer #3:

Remarks to the Author:

General comments

The authors present a method for the classification of nucleotide sequences as according to their natural or synthetic origin. For this purpose, the authors trained a random forest machine learning methodology that achieves 97% accuracy in the predictions using sequence identity and coverage as input features. Furthermore, the authors discuss the current tendencies in applicability of gene synthesis to conclude that new generation technologies are particularly relevant to the scientific community because they succeed at producing longer sequences; a common limitation of other classical methods.

The manuscript is written in adequate English and the ideas are expressed with sufficient clarity to be followed by non-specialist readers.

Major comments

The authors introduce a machine learning-based methodology for the prediction of whether a sequence exists in Nature or is derived from a synthetic construct. The definition of natural and synthetic follows a practical approach, where existing sequences are "natural" and non-existing sequences are "synthetic". After evaluating the features that contribute the most to classification, the authors claim that sequence identity and sequence coverage are the most relevant features. This constitutes logical fallacy; the authors incur in circular reasoning as the fact that sequences were divergent from those in the database is the principle that defines the classification groups.

The triviality of the analysis is also exemplified by the almost perfect (97%) accuracy achieved by a very simplistic classifier. I wonder whether similar results could not have been obtained by using BLASTn directly on the NCBI RefSeq database directly and selecting those beyond the %id cutoff as the one defining the training groups.

Minor comments

An interesting aspect of the manuscript is that the authors discuss current tendencies in the field by analyzing the origin and properties of sequences deposited in the Addgene database. One highlighted result is that the longer a natural sequence is, the less probably it is transferred into another organism. Not sure whether this result should be highlighted so much in the results, as this is a well-established limitation of traditional methods.

Similarly, I would like to see further insight in the discussion regarding the observation that longer phylogenetic distances are observed for synthetic constructs respect to their natural counterparts. I wonder, for example, whether it is a nature of the methods used to incorporate sequences from one type and another. Furthermore, and probably more importantly, I wonder whether the intrinsic definition of synthetic sequence (as distant from anything existing in the database) used in the work is influencing the observation to a high degree. I would intuitively expect that synthetic sequences have a longer distance to any organism (i.e. not only evaluated one).

Authors' Response to Reviewers

We appreciate the constructive feedback shared by reviewers of our manuscript. In response, we have made changes to the manuscript text (all non-trivial changes **highlighted** for ease of identification), and we have responded **in blue** to the reviewer points below.

Reviewers' comments:

Reviewer #1 (Remarks to the Author):

Research presented in this paper works toward the development of methods to detect purposefully engineered organisms (from natural organisms) in the context of biodefense. This is an important need for microbial forensics, and the reason for such programs as FELIX; it is also extremely challenging and the authors have devised a novel means for attempting to do this. The methods here are heavily caveated – and rely on a number of built in assumptions. (For example, use of a dataset which does not likely represent the sampling environment in the context of a bioevent). That said, the ideas here regarding measurable genetic distance (and codon optimization) of synthesized genes vs natural genes, are compelling tools which could be applied to the detection challenge.

We appreciate the enthusiasm about our tools and their applicability to the detection challenge. We agree that testing our methods on the Addgene database does have some drawbacks. The mix of organisms present there over-represents model organisms useful to lab science. Despite that, the Addgene database is informative and valuable. In part, this is because the ideal testing ground – a good publicly-available threat library – does not exist. But beyond this, Addgene's repository contains a broad range of organisms, modified for many purposes, often using cutting-edge modification techniques. Thus, the repository is an excellent testing ground for ensuring that detection techniques work across a broad range of modification types and techniques.

My questions in the biosecurity context relate to the applicability of the tools operationally, as considerations for the authors. The authors reference the production of opioids in yeast, or the production of toxins in E. coli, as examples for biodefense concern, which they are – however these particular examples are not ones for which their analysis would provide the most insight in terms of forensics. Actors who use synthetic biological means to produce illicit drugs or toxins at scale would use the compounds themselves for harm, rather than the microbes that created them (one reason that biological creation of chemicals is so concerning is that many actors have long learned viable means for delivery of chemical toxins, as opposed to the much more challenging delivery of biological pathogens, for example). That said, the discovery of such microbes could possibly occur in forensic investigation, if access to a suspected actor's production facility was available for sampling. The methods being developed by the authors could best be applied to contamination of food, water, or the environment by engineered microbes. In that context the authors may want to speculate on how their methods could be useful when the forensic sample

would necessarily be less pure: coming from soil, mixed aqueous materials, or animal or plant tissues (this is alluded to at the top of page 10 but may warrant further exposition). Similarly the authors may want to note if environmental baselines (assessments of soil, water, etc. for standard organisms and their nucleotide identities) would be useful.

We agree that our examples are chemically-oriented, and that because of this these scenarios today would more likely lend themselves to the direct delivery of the compounds themselves, rather than of the (more challenging delivery) of the biological pathogens themselves. However, we also believe that it is very plausible that actors could use whole organisms. Indeed, one of the authors recently participated in a biological “red teaming” exercise that considered such scenarios. In addition, a report on “Biodefense in the Age of Synthetic Biology” released by the U.S. National Academies in June identified such chemical producer strains as warranting greatest concern, particularly for *in situ* production of biochemicals inside humans. We now cite this report, though we avoided describing such scenarios explicitly in our text (line numbers 81-83).

Our belief that the delivery of whole organisms is a significant threat arises for a number of reasons. Whole organisms have advantages – they can self-replicate, they can persist in environmental populations on the timescale of years (Krumme et al. *Environ Sci Technol* 1997; Ripp et al. *Environ Sci Technol* 2000), and they can be transported covertly if dormant until subsequent induction. Alternatively, an engineered microbe could subtly perturb ecological communities and instigate a secondary effect. We now cite the references above to underscore the potential duration of these effects (lines 350-351).

We agree with the reviewer that an important application of our methodology would be in a forensic context, and that sample purity can complicate identification and classification. If forensic testing were conducted after an incident, one could increase purity by fractionating samples or isolating the troublesome organism based on measuring concentration gradients or conducting other tests. Although we previously alluded to impure samples, we now elaborate upon our treatment of this subject as the reviewer suggested. Specifically, we introduce a proposed workflow for biosecurity-related classification in Supplemental Text and include related discussion in the main text (lines 312-327). This workflow highlights related and complementary approaches for engineered organism determination, such as 16S rDNA amplification/sequencing from populations, to show how our approaches add value to existing tools. Regarding environmental baselines, various communities of microbiome researchers (soil, gut, etc.) have noted that environmental samples can be diverse and dynamic, which makes environmental baseline determination more challenging. But assuming environmental baselines are known, then our workflow notes the context in which they could be used to isolate an “outlier” organism.

The authors have left pathogens out of their study, citing “they are already on watch lists for biosurveillance”. However, pathogens are of high primary interest to microbial forensics – how can authorities discern a naturally occurring outbreak from one that was initiated through an engineered pathogen? Biosurveillance serves a public health requirement – it does not tell us if we are under a biological attack, and the list of diseases for biosurveillance may not cover novel

organisms. Authorities would be unable to attribute such an attack without means to identify that the causative agent had been engineered. It would be helpful for the authors to think about this in the context of developing their tool. For example, would their tool be able to discern if an antibiotic resistance gene were engineered into a common microbe? Or if a fungal species was engineered for temperature resistance and could thus infect humans? That would have immense utility for forensics (similarly, other genes such as those that might confer pathogenicity or other traits in microbes).

When we initially wrote that pathogens are already on watch lists for biosurveillance, we intended to say only that our approach would ignore efforts to resurrect natural pathogens by gene synthesis, which we would be unable to identify as synthetic. However, we agree with the reviewer that engineered pathogens are very important to consider. We mistakenly suggested that they are left out of our study when in fact pathogen genomes are part of the RefSeq database. We verified this by searching for and finding complete genome sequence records of *Clostridium botulinum*, *Corynebacterium diphtheriae*, *Mycobacterium tuberculosis*, *Influenza*, and *Measles*. Thus, synthetic genes that appear in pathogens would be flagged. We now briefly mention that in our discussion (lines 351-353) and have removed our original comment about leaving pathogens out of our study.

The ability of our method to detect the introduction of new antibiotic resistance or temperature resistance would depend on the extent of changes necessary to imbue those traits, and whether they could be imported from other members of the native species. We also limit ourselves to examining open-reading frames for our analysis. This limits our ability to detect modifications to other parts of plasmids or genomes, but it also prevents us from relying too strongly on the exact plasmid presentation for our detection. The implication of this is that if, for example, temperature resistance were conferred via promoter mutations we would be unable to detect them. On the other hand, if temperature resistance were obtained through introduction of one or several non-native genes, then that would exhibit the low sequence similarity within ORFs and non-native source organism maximal-scoring segment pair characteristic of an engineered organism and our method should apply. We've added a sentence clarifying this in our paper (lines 325-327).

I concur that advances in governance of gene synthesis technology can be extremely useful in biosecurity mitigation. I would also note that tools to limit undesired proliferation of engineered organisms is extremely important for biosafety; they would not help, however, in the security context, as they would not be utilized by bad actors.

We believe that molecular approaches for biocontainment, such as synthetic auxotrophy, can be useful for both biosafety and biosecurity (for example, by requiring use of a regulated small molecule for proliferation of an organism engineered with dual use possibilities such that an unlicensed malicious actor could not use it). However, as the reviewer may imply, we think our points about other technology advancements digress from our discussion of sequence classification and we have removed the corresponding sentences.

Other comments

The authors note that the advent of gene synthesis tools are expanding the number and complexity of genes that have been deposited in Addgene. This also likely represents a smaller fraction of what might be more widely occurring in small start ups that are bioprospecting for genes that expressed high value compounds (re sequencing of fungal genomes, soil microbe sampling and sequencing). Whether these genes/gene systems will make their way into shared databases remain to be seen, as some may become proprietary over time.

We agree that the Addgene data represents a small fraction of what may be occurring at start-ups that are bioprospecting. We now mention that in the text to provide context to our readers (lines 92-94).

While plants seem to be in small quantity in Addgene per fig 1D, the authors may want to revisit some of the literature on genomic detection of genetically modified foods/crops/seeds to determine if their tool has potentially improved utility in contrast to methods currently used in the agriculture realm.

We thank the reviewer for bringing this relevant set of literature to our attention and agree that our tools may indeed be improvements on those used in agriculture. The literature indicates that methods currently used in agriculture rely on amplification of known genes associated with common GMOs. Our revised Discussion has a new paragraph treating this subject along with relevant references (lines 328-345).

Small typo:

Top of page 4, I believe the word “repositories” should be ‘repository”.

I hope these comments are useful – thank you for the opportunity to provide a biosecurity review.

Indeed. Fixed. Thanks for the careful read.

Reviewer #2 (Remarks to the Author):

In manuscript NCOMMS-18-01285A-Z, Thompson et al. describe an algorithm to identify synthetic DNA sequences from natural gene sequences. In the second part of the manuscript, they apply this algorithm to sequences extracted from Addgene database. This analysis allows them to identify major trends in the way gene synthesis is used by the life science community. The results of each section are interesting and timely. The criteria used to identify synthetic genes are sophisticated and appear to be very effective. The conclusions of the Addgene analysis are interesting if not completely unexpected.

Major Revisions

1- The focus of this paper seems to be shifting between the introduction and the conclusion. The introduction is mostly focused on bio-surveillance and the identification of synthetically engineered organisms. The conclusion is mostly focused on analyzing trends in the use of gene synthesis. In my opinion, these are two different questions and I am not sure it makes sense to bundle them in the same manuscript.

We agree that there are two topics of focus in this manuscript, our method for detecting synthetic genes and applications of the method. We discuss two applications, one focused on trends in the use of gene synthesis (which is what we used the method for in this paper) and the other focused on bio-surveillance (which is likely to be the primary utility of our approach in future studies). Based on constructive reviewer feedback, we have more effectively tied these perspectives together in our revised narrative, particularly in the introduction and discussion. In short, our introduction now includes just one paragraph focused on biosurveillance opportunities to provide motivation in this direction, whereas our discussion dedicates several forward paragraphs to the consequences and utility of our approaches for biosurveillance.

2- The classification of genes in synthetics vs. not synthetic is fine for analyzing the Addgene repository but it is very underdeveloped to claim it would be useful in bio-surveillance. There is no critical analysis of its limitations or its benefits compared to other methods. For instance, the algorithm proposed by the government to screen synthetic DNA orders (doi:10.1038/nbt.1802) would be able to detect that different segments of a large DNA sequence are associated with different species. I am not convinced that the codon optimization criteria proposed by the author would result in a major gain of performance. In order to claim that this approach can be used in security applications, I think that the performance of the algorithm would need to be characterized much more thoroughly. It would need to go beyond simply detecting that a gene has been synthesized commercially.

Thanks for highlighting this other exciting stream of work. We agree that there are connections between this stream and identification of synthetic genes in organisms, and we now reference more of literature on screening synthetic DNA orders in our introduction (lines 72-77). However, we also contend that there are important differences. One particularly important difference in the

context that it is used: analyzing sequences derived from organisms found in labs or in the environment, rather than when synthetic DNA is ordered.

As another of our reviewers with biosecurity expertise noted, there is significant interest in understanding whether a potentially threatening organism has been engineered. This question is a foregone conclusion in the ordering context, but for our application it is crucial. For this reason, we contend that the codon-optimization and genetic distance classification methods do add value beyond those of algorithms to screen synthetic DNA orders. Our addition of a proposed workflow for engineered organism detection in the Supplement and related discussion (lines 307-322) further showcase the differing context in which our approaches would be used compared to those at the DNA ordering stage.

3- In order to address 1 and 2, I would suggest rewriting the introduction to deemphasize the bio-surveillance aspect and simply introduce the classification algorithm in the context of the Addgene analysis.

Because reviewers felt as though our bio-surveillance claims could be strengthened, we have bolstered the description of how our methods contribute to bio-surveillance applications. We make several improvements to the biosecurity relevance, including addressing operational considerations by including a proposed workflow in the Supplement.

Nevertheless, to further improve the paper and to address your concerns we have rewritten the introduction to achieve better balance and integration between these two aspects of the paper. As mentioned earlier, the introduction now only features one paragraph pertaining to biosurveillance (lines 72-83).

4- The first paragraph of page 5 is not very clear. My understanding from the figure is that analysis is performed at the gene level but this is not stated anywhere. Also, it is not clear if the analysis is performed based on sequence annotations or for all open reading frames. Finally, I don't understand why the authors use a nucleotide BLAST search to identify the source organisms. It seems to me that they would get better results by doing a BLAST on the translated DNA sequence as protein sequences would be unaffected by codon optimization.

We thank the reviewer for raising several thoughtful points here that improve our manuscript. Although we did state in our Online Methods (contained within the Supplement) that the analysis is of ~19,000 unique ORFs from the Addgene database, we did not carefully indicate how ORFs were determined. We now include more details on how Addgene determined the ORFs that we used.

The choice of BLASTn or BLASTx for source organism determination is an interesting one that we considered carefully at the outset, but our initial explanation in the Supplement did not reflect this deliberation. We now explain several reasons why we chose BLASTn. In short, it would not require an additional BLAST run of each queried sequence, and we believe it may generally be more accurate for source organism determination. This is because protein collections are more

sparsely populated than nucleotide collections and because proteins are more highly conserved across organisms, muddying the water for the source organism. For example, we analyzed 50 randomly obtained sequences from the Addgene dataset and found ones that resulted in maximal-scoring segment pairs of 100% query coverage and 100% sequence identity using BLASTn but resulted in maximal-scoring segment pairs of 100% query coverage and 99% sequence identity using BLASTx (and a different organism listed). Rather than describe this result, in our Supplement we examined whether these methods predict different phyla for the source organisms. As described in our revised Supplement, in only 5 of the 50 cases did the phyla differ. For the 5 that do differ, we do not have evidence that BLASTn or BLASTx is more accurate. Thus, with no evidence that BLASTx performs better and the known drawbacks of a smaller reference database and the potential for greater organism confusion, we chose BLASTn.

5- The data used in this report should be included in the online supplement or deposited in a public repository such as Figshare. The tables S4 and onward are not useful in their current form. I would like to see the sequences of these genes made available in a computer-readable format. It would also be very interesting to have access to the Addgene primary data. The data should be accompanied by an open data license.

We are happy to make as much of our data more accessible as possible. We have now included machine-readable version of S4-S11 for the online supplement, along with an open data license statement. The Addgene repository data, however, is proprietary to them and thus we are unable to provide access to it.

Minor Revisions

1- I think that citing a review about gene synthesis would be appropriate in the second paragraph of the introduction. I suggest [10.1016/j.tibtech.2008.10.007](https://doi.org/10.1016/j.tibtech.2008.10.007) and [doi:10.1038/nmeth.2918](https://doi.org/10.1038/nmeth.2918)

We agree with the reviewer and now cite both these reviews in our introduction (line 52).

2- The supplement includes the scripts used in this work. They would be more useable if provided separate text files. I would also encourage the inclusion of an OSI-approved open source license file in the directory containing the scripts.

The authors appreciate all the reviewer suggestions that facilitate community access to our tools. We now supply the scripts and an accompanying license file.

Jean Peccoud

jean.peccoud@colostate.edu

Reviewer #3 (Remarks to the Author):

General comments

The authors present a method for the classification of nucleotide sequences as according to their natural or synthetic origin. For this purpose, the authors trained a random forest machine learning methodology that achieves 97% accuracy in the predictions using sequence identity and coverage as input features. Furthermore, the authors discuss the current tendencies in applicability of gene synthesis to conclude that new generation technologies are particularly relevant to the scientific community because they succeed at producing longer sequences; a common limitation of other classical methods.

The manuscript is written in adequate English and the ideas are expressed with sufficient clarity to be followed by non-specialist readers.

Major comments

The authors introduce a machine learning-based methodology for the prediction of whether a sequence exists in Nature or is derived from a synthetic construct. The definition of natural and synthetic follows a practical approach, where existing sequences are “natural” and non-existing sequences are “synthetic”. After evaluating the features that contribute the most to classification, the authors claim that sequence identity and sequence coverage are the most relevant features. This constitutes logical fallacy; the authors incur in circular reasoning as the fact that sequences were divergent from those in the database is the principle that defines the classification groups.

The triviality of the analysis is also exemplified by the almost perfect (97%) accuracy achieved by a very simplistic classifier.

We regret that it came across that our argument was circular. It definitely is not, as we explain below, but in re-reading our description we can see how this might have been conveyed by what we wrote. We’ve now revised the text (lines 121-123, 166-169) and Figure 2 to clarify this.

To the substantive critique of circularity, the first important point is that our ‘true’ labels for both our training and test sets were not determined by inclusion / exclusion in reference databases, but by finding sequences in the scientific literature that were known to be natural or synthetic (as described on lines 166-169). The connection to the databases comes when we form our ‘predicted’ labels for each sequence. Based on the similarity of a target sequence to the databases we form a hypothesis as to whether it is synthetic. Thus our predicted labels are based on the databases, whereas our true labels are based on references to the underlying science, and hence there is no circularity to our findings. The 97.7% success rate simply reflects that our technique is highly predictive.

We now also show that our determination of the quantitative 85% sequence identity threshold is not trivial. In the Supplemental data we now show how other potential thresholds (75%, 80%,

90%, and 95%) perform worse on our test data set than our chosen threshold (Supplemental Table 12).

I wonder whether similar results could not have been obtained by using BLASTn directly on the NCBI RefSeq database directly and selecting those beyond the %id cutoff as the one defining the training groups.

While our process was not circular (as outlined above), this alternative would be. By using NCBI RefSeq %id as the definition of the training groups, the analysis would then become trying to predict NCBI RefSeq %id using RefSeq %id. This would, of course, be 100% predictive, but still completely uninformative, because it wouldn't be predicting whether something was actually created synthetically, but just a heuristic.

Moreover, using this for the training set would preclude testing for the importance of many other potential factors that could help determine whether a sequence was synthetic, because the RefSeq %id variable would already be (incorrectly) explaining 100% of the variance (because it was predicting against incorrect labels). This would have prevented us from exploring many other plausible hypotheses for ways to detect synthetic sequences (GC content, rare codon usage, etc.).

Finally, this would be using an empirically-derived method to predict a heuristic (not the real world). Because of this, it would be uninformative about the underlying question of interest: whether the sequence truly was engineered. As such, this procedure could not provide any calibration numbers (equivalent to our 97.7% above) about its true predictive power.

It is thus crucial to both the science underpinning our method and our confidence that the result would be usefully predictive in the real world of 'true' labels that our training set not be defined using a BLASTn cutoff in the NCBI RefSeq database.

Minor comments

An interesting aspect of the manuscript is that the authors discuss current tendencies in the field by analyzing the origin and properties of sequences deposited in the Addgene database. One highlighted result is that the longer a natural sequence is, the less probably it is transferred into another organism. Not sure whether this result should be highlighted so much in the results, as this is a well-established limitation of traditional methods.

We agree that the result that the longer a *natural* gene sequence is, the less likely it is to be transferred into another organism by researchers is consistent with the perception that longer unmodified gene sequences are generally more difficult to express and that, as sequence length grows, so does the likelihood that there will be sequence regions that are troublesome to express in another organism. (Although we do feel that it is a contribution to show this empirically.)

However, our broader intention in highlighting this pattern in natural sequences is to draw the contrast with synthetic sequences, where there is little to no drop in genetic distance as gene length grows, and at large lengths are used predominantly for transfer across distant organisms.

We also note that the discussion of natural sequence length and propensity for transfer in the original manuscript is limited to only two sentences, one of which acknowledges that the result is expected. Thus we feel this result is not highlighted excessively.

Similarly, I would like to see further insight in the discussion regarding the observation that longer phylogenetic distances are observed for synthetic constructs respect to their natural counterparts. I wonder, for example, whether it is a nature of the methods used to incorporate sequences from one type and another. Furthermore, and probably more importantly, I wonder whether the intrinsic definition of synthetic sequence (as distant from anything existing in the database) used in the work is influencing the observation to a high degree. I would intuitively expect that synthetic sequences have a longer distance to any organism (i.e. not only evaluated one).

We understand the intuition behind this concern, but this is not what is occurring. This can be seen both in practice and in theory.

In practice, it is true that our definition of a synthetic sequence does mean that there are changes in the nucleotide sequence. But this distance is normalized by total sequence length to produce a percentage, and thus is not influenced by that length. And, as can be seen in Figure 4, synthetic sequences are actually closer at small sizes showing that synthetic parts are not intrinsically further away.

In theory this isn't a concern because our measurements of phylogenetic distances are independent of the number of nucleotide changes. In particular, the BLAST search (on either a natural or synthetic sequence) yields the closest natural organism (our "source" organism). We use that organism's 16S ribosomal RNA as the source irrespective of whether the sequence searched was synthetic or natural. Similarly, the expression organism comes from an Addgene notation of how it was actually used. Hence, in both cases we are measuring distances from one natural organism to another, and thus the 'synthetic' aspect cannot be biasing the result. We now provide clarifying text related to our phylogenetic distance calculations in the Results section (lines 194-196, 219-222).